# Combination of EphA2- and Wee1-Targeted Therapies in Endometrial Cancer

**DOI:** 10.3390/ijms24043915

**Published:** 2023-02-15

**Authors:** Santosh K. Dasari, Robiya Joseph, Sujanitha Umamaheswaran, Lingegowda S. Mangala, Emine Bayraktar, Cristian Rodriguez-Aguayo, Yutuan Wu, Nghi Nguyen, Reid T. Powell, Mary Sobieski, Yuan Liu, Mamur A. Chowdhury, Paola Amero, Clifford Stephan, Gabriel Lopez-Berestein, Shannon N. Westin, Anil K. Sood

**Affiliations:** 1Department of Gynecologic Oncology and Reproductive Medicine, The University of Texas MD Anderson Cancer Center, Houston, TX 77030, USA; 2National Institute of Animal Biotechnology, Hyderabad 500029, India; 3UTHealth Houston Graduate School of Biomedical Sciences, The University of Texas MD Anderson Cancer Center, Houston, TX 77030, USA; 4Department of Experimental Therapeutics, The University of Texas MD Anderson Cancer Center, Houston, TX 77030, USA; 5High-Throughput Research and Screening Center, Center for Translational Cancer Research, Texas A&M Health Science Center, Institute of Biosciences and Technology, Houston, TX 77030, USA

**Keywords:** endometrial cancer, EphA2, Wee1

## Abstract

EphA2 tyrosine kinase is upregulated in many cancers and correlated with poor survival of patients, including those with endometrial cancer. EphA2-targeted drugs have shown modest clinical benefit. To improve the therapeutic response to such drugs, we performed a high-throughput chemical screen to discover novel synergistic partners for EphA2-targeted therapeutics. Our screen identified the Wee1 kinase inhibitor, MK1775, as a synergistic partner to EphA2, and this finding was confirmed using both in vitro and in vivo experiments. We hypothesized that Wee1 inhibition would sensitize cells to EphA2-targeted therapy. Combination treatment decreased cell viability, induced apoptosis, and reduced clonogenic potential in endometrial cancer cell lines. In vivo Hec1A and Ishikawa-Luc orthotopic mouse models of endometrial cancer showed greater anti-tumor responses to combination treatment than to either monotherapy. RNASeq analysis highlighted reduced cell proliferation and defective DNA damage response pathways as potential mediators of the combination’s effects. In conclusion, our preclinical findings indicate that Wee1 inhibition can enhance the response to EphA2-targeted therapeutics in endometrial cancer; this strategy thus warrants further development.

## 1. Introduction

EphA2 is a receptor tyrosine kinase that has multiple roles in facilitating malignant progression. Although EphA2 was first studied in the context of neuronal migration during embryogenesis, it has since been shown to regulate cancer cell growth, migration, invasion, and angiogenesis [1]. In addition, EphA2 is overexpressed in various cancers, including breast cancer [2], esophageal cancer [3], melanoma [4], lung cancer [5], prostate cancer [6], ovarian cancer [7,8], and endometrial cancer [9,10]. Over the years, many therapeutic strategies have been developed to target EphA2, including tyrosine kinase inhibitors, monoclonal antibodies, immunoconjugates, aptamers, and short-interfering RNA (siRNA) [11,12]. We have previously demonstrated that delivery of EphA2 siRNA through 1,2-dioleoyl-*sn*-glycero-3-phosphatidylcholine (DOPC) neutral liposome nanoparticles (EPHARNA) showed highly efficient in vivo delivery to the tumor, resulting in decreased tumor burden in mouse ovarian cancer models [13]. In addition, EPHARNA combined well with paclitaxel and significantly reduced tumor growth in these preclinical models [13]. A phase 1 trial of EPHARNA in patients with solid cancers is ongoing. However, rational combinations with EphA2-targeted therapy are not yet known; therefore, we aimed to identify novel therapeutic combinations through high-throughput chemical screens for use in endometrial cancer.

In the present study, we identified the Wee1 kinase inhibitor, MK1775, as a synergistic partner to EphA2-targeted therapy in endometrial cancer cells. To test our hypothesis that Wee1 inhibition sensitizes cells to EphA2-targeted therapy, we examined the anti-tumor effects of both agents in endometrial cancer mouse models and evaluated potential mechanisms of synergy. Our findings supported our hypothesis, justifying further investigation of this combination.

## 2. Results

### 2.1. High-Throughput Drug Screening Identifies Rational Combinations to EphA2 Inhibition

Given the oncogenic function of EphA2 in endometrial cancer, we performed a systematic high-throughput drug screen on an endometrial cancer cell line (Hec1A) transfected with either control siRNA or EphA2 siRNA with two drug libraries containing a total of 1510 drugs comprising FDA-approved drugs, clinical candidates, active pharmaceutical ingredients, and chemotherapeutic agents. We hypothesized that candidate small molecule drugs from these libraries would overcome resistance to anti-EphA2 therapeutics such as EPHARNA and thereby enhance their therapeutic benefit when given as combined therapy. Among the top hits, the biggest class represented was targeted kinase inhibitors (MK-1775, TAK-632, and BML-277), followed by microtubule poisons (darinaparsin and CYT997) and GPCR and G protein inhibitors (azilsartan, medoxomil, and matrine) (Figure 1A). Among the three kinase inhibitors in the hit list, the Wee1 inhibitor, MK1775, demonstrated the greatest synergistic interaction score (Bliss synergy score of 1.14) when combined with the EphA2 inhibitor, ALW (Figure 1C,D). Furthermore, EphA2 knockdown increased Wee1 activity as evidenced by elevated phospho-cdc2 levels (Figure 1B), hinting that Wee1 activation may be a compensatory mechanism to adapt to the loss of EphA2 and maintain cell viability.

Next, we analyzed the synergism of EphA2 and Wee1 inhibition in two endometrial cancer cell lines (Hec1A and Ishikawa) known to have high expression levels of EphA2. To determine the effect of combination therapy of EphA2 and Wee1 inhibitors (ALW and MK1775, respectively), cancer cells were treated with either each single drug or both in combination for a period of 72 h, followed by cell viability analysis using an MTT assay. We observed a significant decrease in cell viability in a dose-dependent manner in both Hec1A and Ishikawa cells; at each dose tested, cell viability was lower after combination therapy than after individual drug treatments (Appendix A). Furthermore, we used SynergyFinder and CompuSyn software to test the drug–drug interactions using the median effect equation to derive combination index values, and we observed that the drug combination produced a synergistic effect (Figure 1E,F).

### 2.2. MK1775 Sensitizes Endometrial Cancer Cells to EphA2 Inhibition

Upon morphological analyses, single-drug-treated wells showed a lower number of cells in comparison to control cells, and the cell numbers were even lower with the combination treatment (Figure 2A). Of interest here is the appearance of many round cells for the ALW and MK1775 combination treatment at both the 24 h and 48 h time points, which could be either cells undergoing mitosis or cells preparing to die by apoptosis. To understand this, we performed cell cycle analysis and observed that there was a decrease in number of cells in the G1 phase and a concomitant increase in number of cells in the S phase for the combination treatment at the 48 h time point (Figure 2B and Appendix A). In addition, there was a statistically significant increase in the sub-G1 population in cells given combination treatment compared with the control cells (Figure 2C). Consistent with the associated changes in the cell cycle profile (Figure 2D and Appendix A), we observed elevated levels of the mitotic marker phospho H3 (Ser10) in cells treated with the drug combination compared with cells treated with single drugs or the control cells (Figure 2E). This result indicates that the cells moved into mitosis prematurely, overriding the G2/M checkpoint (Figure 2D) and thereby triggering mitotic arrest and leading to mitotic-catastrophe-mediated apoptotic cell death.

The synergistic effect of the combination treatment was further evaluated using an Annexin V/PI-based apoptosis assay. Higher levels of apoptosis were seen for combination treatment with ALW and MK1775 than for individual drug treatments in both cell lines tested (Figure 3A,B). This result was confirmed using PARP cleavage as a marker of apoptotic cell death; as expected, we saw higher levels of cleaved PARP in the combination treatment (Figure 3C,D). In addition to caspase-3 activation (Appendix A) and PARP cleavage, one of the hallmarks of apoptosis is double-strand DNA break formation, which is marked by γH2AX phosphorylation at S139. We observed increased levels of γH2AX phosphorylation after combination treatment, indicating that extensive double-strand breaks occurred during cell death (Appendix A). Furthermore, we performed a colony formation assay to assess the effect of therapy on clonogenic survival, and in comparison to control cells, cells treated with the drug combination had significantly suppressed colony-forming activity in Hec1A (*p* < 0.001) and Ishikawa (*p* < 0.01) cells (Figure 3E,F).

### 2.3. EPHARNA and MK1775 Suppress Tumor Growth in an Endometrial Cancer Xenograft Model

To test the anti-tumor effects of EPHARNA and MK1775 in vivo, we next used the Hec1A and Ishikawa-Luc orthotopic endometrial cancer models. Tumor nodules were localized primarily in the uterus with few metastases to the intestine, stomach, and peritoneal wall (Figure 4A,E). At the end of the experiment, mice treated with EPHARNA and MK1775 monotherapies showed lower tumor weight (Figure 4B) and fewer tumor nodules (Figure 4C) compared with the control group, although the differences did not reach statistical significance. In contrast, EPHARNA and MK1775 combination therapy led to significantly lower tumor weight (*p* < 0.05) and fewer tumor nodules compared with the control group (Figure 4B,C). There were no significant differences in mouse body weight across all four groups (Figure 4D).

In the Ishikawa-Luc model, treatment with EPHARNA and MK1775 combination therapy resulted in a significant decrease in tumor burden in comparison to siControl or EPHARNA monotherapy (Figure 4E,F). However, we observed higher ascites in siControl-, EPHARNA-, and MK1775-treated animals, which is partly attributed to the increased body weight in siControl- and EPHARNA-treated groups in comparison to combination treatment (Figure 4G,H).

### 2.4. Inhibition of EphA2 and Wee1 Leads to Suppression of DNA Damage Response Repair Pathways and Downregulates Cell Survival Pathways

To identify potential downstream mechanisms associated with the synergistic interaction of EphA2 and Wee1 inhibition, we performed RNA-Seq analysis followed by IPA analysis. To better analyze the observed synergy between the drugs, we performed a comparative analysis to identify enriched canonical pathways that were high under EphA2 inhibition alone and low under combination therapy. This analysis identified therapeutic compensatory cell survival pathways activated under EphA2 monotherapy that were lost under combination therapy. Comparative analysis identified downregulation of cell cycle checkpoint regulation, DNA damage response, and cell survival pathways (Figure 5A).

Further investigation was performed to better understand the molecular mechanisms associated with the synergism of Epha2 and Wee1 inhibition. In comparison to monotherapy, pAKT levels were lower with the combination therapy, hinting at reduced cell survival capacity (Figure 5B). In addition, an increase in phosphorylated S6 levels served as a surrogate for mTOR activation status (Figure 5C). Because mTOR was activated with the combination treatment, we examined the status of autophagy in these cells and observed that the combination treatment led to a block in the autophagic flux, as evidenced by the accumulation of LC3B-II and p62 levels (Figure 5D,E). These results suggest that presence of a block in autophagic flux, along with additional defects in cell cycle progression and DNA damage responses, pushes the cells into apoptotic cell death.

## 3. Discussion

In this study, we sought to identify novel agents for combination with EphA2-targeted therapy in endometrial cancer. One of the top hits identified in our high-throughput screen was the Wee1 kinase inhibitor, MK1775. We found EphA2- and Wee1-targeted therapies had synergistic effects in vitro, and the combination therapy led to enhanced anti-tumor efficacy in vivo.

Over two decades of preclinical research have identified EphA2 as a promising target for clinical translation. Previous studies have shown that EphA2 silencing was effective in reducing tumor burden in combination with chemotherapeutic agents such as paclitaxel and docetaxel [13]. Here, we showed that silencing EphA2 expression with siRNA increases Wee1 activity, as evidenced by increased cdc2 phosphorylation, which regulates the G2/M cell cycle block and allows cells to evade cell death by inhibiting premature entry into mitosis. Therefore, increased Wee1 activity may serve as an adaptive cell survival mechanism to evade cell death upon EphA2 inhibition. Blocking such secondary cell survival loops shows promise in enhancing the therapeutic efficacy of EphA2 inhibition.

In vitro experiments confirmed the synergy between the EphA2- and Wee1-targeted drugs, as seen through an increase in apoptosis and the inhibition of colony-forming efficiency. Furthermore, RNA-Seq analysis confirmed the synergy was partly mediated by suppression of cell proliferation and of DNA damage response pathways. EphA2 has been shown to be highly expressed in many cancers and has been shown to regulate the PI3K-AKT signaling pathway. However, it has been shown to have opposing effects in different cancers. For example, in pancreatic cancer and hepatocellular carcinoma, ligand-induced EphA2 signaling activates AKT signaling by enhancing its phosphorylation [14,15]. In contrast, in glioblastoma, EphA2 signaling decreases AKT function by reduced phosphorylation [16]. Here, we observed that EphA2 inhibition decreased AKT signaling modestly, with further reduction when EphA2 inhibition was combined with the Wee1 inhibitor MK1775. This suppression of AKT signaling with the combination treatment enhanced mTOR activity, leading to inhibition of cell survival autophagy and ultimately leading to cell death by apoptosis in the absence of efficient DNA damage response activation.

At present, several EphA2-targeted therapeutics are being evaluated in clinical trials in various cancers in which EphA2 has an established oncogenic function [10,12]. Our findings provide further support to explore additional combination therapies that may synergize with EphA2-targeted therapeutics. In addition, recent ADAGIO phase 2 trial results of the Wee1 inhibitor adavosertib (AZD1775) demonstrating response rates of 30% in uterine serous carcinoma are promising and warrant further investigation [17]. Furthermore, since EphA2 is overexpressed in ovarian serous carcinoma and is associated with poor clinical outcomes [18], it is possible that the combination therapy with Epha2 inhibition may be a viable option for further improving clinical outcomes of patients with other histological subtypes such as serous carcinoma.

## 4. Materials and Methods

### 4.1. Cell Culture

Hec1A (RRID:CVCL_0293) and Ishikawa cells (RRID:CVCL_2529) were procured from ATCC and The University of Texas MD Anderson Cancer Center Characterized Cell Line Core, respectively. Cell lines were validated by short tandem repeat fingerprinting in the core facility. Cells were routinely screened for mycoplasma. Hec1A cells were grown in McCoy’s 5A medium (HyClone, Logan, UT, USA), and Ishikawa cells were grown in Dulbecco’s modified Eagle’s medium (HyClone, Logan, UT, USA), supplemented with 10% fetal bovine serum (Sigma-Aldrich, St. Louis, MO, USA) and 0.1% gentamicin sulfate (Gemini Bioproducts, West Sacramento, CA, USA). Cells were incubated in a humidified atmosphere containing 5% CO_2_ at 37 °C. All experiments were conducted with cells at 70% to 80% confluence and cultured for fewer than 20 passages for in vitro work and for fewer than 10 passages for in vivo experiments.

### 4.2. siRNA Transfection

Cells were plated in six-well plates so that the cells could reach 60% to 70% confluence by the next day. For each well, 1.3 µg of siRNA was added to 150 µL of reduced serum medium (Opti-MEM, Thermo Fisher Scientific, Waltham, MA, USA), and in a separate tube, 8 µL of Lipofectamine RNAiMAX transfection reagent (Thermo Fisher Scientific, Waltham, MA, USA) was incubated in 150 µL of Opti-MEM for 5 min. The siRNA/media mixture was added dropwise to the transfection reagent mixture, vortexed and then incubated for 15 to 20 min at room temperature. Wells to be transfected were washed once with PBS, and then 900 µL of Opti-MEM and 300 µL of siRNA mixture were added dropwise to each well. The plates were gently swirled and placed in the incubator for 4 to 6 h. The transfection medium was replaced with complete media. For the high-throughput screening, after 24 h of transfection, cells were trypsinized, counted, and seeded in clear-bottom 384-well plates. For Western blot analysis, cells were collected 48 to 72 h after transfection.

### 4.3. High-Throughput Screening

High-throughput chemical screens were performed by the Gulf Coast Consortia’s Combinatorial Drug Discovery Program at the Institute of Biosciences and Technology in Texas A&M Health Science Center, Houston, TX, USA. Hec1A endometrial cancer cells transfected with either siControl or siEphA2 were screened against two drug library collections: The Broad Collection-Informer Set (358 compounds) and Selleck Bioactives Collection (1150 compounds) libraries. A brief description of the contents of the libraries can be found at the following link (https://ibt.tamu.edu/cores/high-throughput/core-libraries/approved-drugs.html, accessed on 12 July 2019). For screening assays, a total of 800 cells per well were suspended in 50 µL of media and seeded into black 384-well µClear plates (Greiner Bio-One International, Monroe, NC, USA) using a Multidrop Combi liquid dispenser (Thermo Fisher Scientific, Waltham, MA, USA). After cell seeding, the plates were kept at room temperature for 40 to 60 min before being moved into a cell culture incubator. The cells were grown overnight at 37 °C in a humidified chamber (>95% relative humidity) with 5% CO_2_. The following day, 50 nl of drugs were transferred into each well using an Echo 550 acoustic dispensing platform (Labcyte, San Jose, CA, USA). A non-treated plate was immediately fixed with 4% paraformaldehyde and followed by nuclei staining with 4′,6-diamidino-2-phenylindole (DAPI) at the start of drug treatment (day 0) to estimate the number of cells present at the start of treatment. 

In the primary screen, the drug libraries were tested at three concentrations (1 µM, 0.1 µM, and 0.01 µM) with a fixed volume of dimethyl sulfoxide (DMSO) (0.1% *v*/*v*) and two biological replicates. Each assay plate contained a fixed concentration of the drugs in addition to a negative control (0.1% DMSO) and two positive controls (etoposide and dasatinib). After 72 h of incubation, plates were fixed with 0.4% paraformaldehyde and nuclei stained with DAPI using an integrated HydroSpeed plate washer (Tecan Life Sciences, Männedorf, Switzerland) and Multidrop Combi dispenser. Plates were imaged on an IN Cell Analyzer 6000 laser-based confocal imaging platform (GE Healthcare Bio-Sciences, Marlborough, MA, USA), and nuclei were counted using IN Cell Developer Toolbox software (version 1.6). To evaluate the cells’ response to the drug screen, we performed curve fitting followed by a calculation of area under the curve values.

### 4.4. Orthogonal Cell Viability Assay

To evaluate the cytotoxicity of ALW-II-41-27 (ALW; ApexBio Technology, Houston, TX, USA) and MK1775 (ApexBio Technology, Houston, TX, USA) both alone and in combination, cells were plated in a 96-well plate at a starting density of 2000 cells per well for Hec1A cells and 1000 cells per well for Ishikawa cells. After 24 h, the medium was aspirated, and 100 µL of fresh medium containing serial dilutions of individual drugs was placed over the cells. After 72 h of incubation, the medium was aspirated, and cells were incubated with 0.05% MTT solution for 1 h. The supernatant was removed, and the formazan crystals were dissolved in 100 µL DMSO. The plates were read at 570 nm by a uQuant microplate spectrophotometer (BioTek, Winooski, VT, USA). Triplicate biological experiments were performed. Dose–response curves were plotted using Prism 8.0.0 (GraphPad Software, San Diego, CA, USA), the combination index was determined by CompuSyn software [19] (ComboSyn, combosyn.com, accessed on 12 July 2019), and synergy assessment was performed using the Bliss model in the SynergyFinder web application [20] (https://synergyfinder.fimm.fi, accessed on 29 July 2019). The Bliss synergy scores in this platform indicate synergy if they are greater than 10, additivity if they are between −10 and 10, and antagonism if they are less than −10. Based on the cell viability results for all future experiments, Hec1A cells were treated with DMSO, 1 µM ALW, 0.5 µM MK1775, and the combination of 1 µM ALW and 0.5 µM MK1775. Ishikawa cells were treated with DMSO, 0.5 µM ALW, 0.25 µM MK1775, and the combination of 0.5 µM ALW and 0.25 µM MK1775.

### 4.5. Cell Cycle and Apoptosis Analysis

For the cell cycle assay, control and drug-treated cells were trypsinized, washed with PBS twice, fixed in ice-cold 70% ethanol, and stored at –20 °C. On the day of analysis, cells were washed twice with PBS and then incubated in 50 µg/mL of propidium iodide (PI) solution containing 0.5 µg/mL RNase A for 4 h in the dark and analyzed by flow cytometry. For the apoptosis assay, the cell supernatant as well as trypsinized cells were mixed and pelleted and then washed with PBS. The apoptosis assay was performed using the FITC Annexin V Apoptosis Detection Kit I (BD Biosciences, Franklin Lakes, NJ, USA). After Annexin V–fluorescein isothiocyanate (FITC) and PI staining, cells were analyzed by flow cytometry.

### 4.6. Colony Formation Assay

Cells were seeded at a density of 500 to 1000 cells per well in a 12-well plate, and the cells were left to grow in the incubator for 10 to 14 days with the respective drug combinations for both Hec1A and Ishikawa cells. After visible colonies containing more than 50 cells appeared, the plates were fixed with a solution containing glutaraldehyde (6.0%, *v*/*v*) and crystal violet (0.5%, *w*/*v*) for 15 to 20 min at room temperature. After that, the crystal-violet-fixing solution was decanted, and the plates were washed in water three to five times and then left to dry at room temperature. The plates were imaged, and the number of colonies was counted.

### 4.7. Western Blotting

Harvested cells were spun down at 2000 rpm for 5 min, washed with ice-cold PBS, and then pelleted at 3000 rpm for 3 min. Cell pellets were lysed in RIPA buffer supplemented with protease and phosphatase inhibitors and quantified using a Pierce BCA protein assay kit (Thermo Fisher Scientific, Waltham, MA, USA). Equal amounts of protein (20 µg) were boiled at 95 °C for 10 min, run on an SDS-PAGE gel (8–12%), transferred onto a nitrocellulose membrane, incubated in 5% milk (in Tris buffered saline-Tween 20 [TBS-T]) for 1 h, and then incubated overnight in the appropriate primary antibodies (listed below). Blots were washed with TBS-T thrice for 5 min each and then incubated with corresponding secondary antibodies (1:2500 dilution, GE Healthcare, Chicago, IL, USA) for 1 h. Enhanced chemiluminescence substrate (ECL; Thermo Fisher Scientific, Waltham, MA, USA) was then added to the blots for 1 min, and immunoblot images were captured using an Azure Biosystems imaging machine (Azure Biosystems, Dublin, CA, USA).

The following antibodies were used: anti-EphA2 (1:1000 dilution; Cell Signaling Technology, Danvers, MA, USA), anti-phosphorylated (phospho) cdc2 (1:1000 dilution; Cell Signaling Technology, Danvers, MA, USA), anti-cdc2 (1:1000 dilution; Cell Signaling Technology, Danvers, MA, USA), Cell Cycle and Apoptosis WB Cocktail (pCdk/pHH3/Actin/cleaved PARP) (1:250 dilution; Abcam, Cambridge, UK); Apoptosis and DNA damage WB Cocktail (pH2A.X/GAPDH/cleaved PARP) (1:250 dilution; Abcam, Cambridge, UK); anti-phospho S6 (1:1000 dilution; Cell Signaling Technology, Danvers, MA, USA), anti-S6 (1:1000 dilution; Cell Signaling Technology, Danvers, MA, USA), anti-AKT (1:1000 dilution; Cell Signaling Technology, Danvers, MA, USA); anti-pAKT (1:1000 dilution; Cell Signaling Technology, Danvers, MA, USA); anti-cleaved caspase 3 (1:1000 dilution; Cell Signaling Technology, Danvers, MA, USA); anti-P62 (1:3000; BD Biosciences, Franklin Lakes, NJ, USA); anti-GAPDH (1:5000; Thermo Fisher Scientific, Waltham, MA, USA), anti-LC3B (1:1000 dilution; Cell Signaling Technology, Danvers, MA, USA), anti-alpha-tubulin (1:1000 dilution; Cell Signaling Technology, Danvers, MA, USA), and antibeta-actin (1:3000; Sigma-Aldrich, St. Louis, MO, USA).

### 4.8. Liposomal Nanoparticle Preparation

For in vivo delivery, siRNAs were incorporated into DOPC liposomes as described earlier [13]. In brief, DOPC and siRNA were mixed in a ratio of 1:10 (*w*/*w*) siRNA:DOPC in the presence of excess tertiary butanol. Tween 20 was added to the siRNA/DOPC mixture in a ratio of 1:19 (Tween-20:siRNA/DOPC). The mixture was vortexed, frozen in an acetone/dry-ice bath, and lyophilized. Before in vivo administration, this preparation was hydrated with magnesium- and calcium-free PBS to achieve a desired concentration of 5 µg of siRNA in 200 µL volume per dose per mouse.

### 4.9. In Vivo Model of Endometrial Cancer

Female nude mice aged 4–8 weeks were purchased from Taconic Biosciences, USA. All mice were housed at The University of Texas MD Anderson Cancer Center animal facility under specific pathogen-free conditions. All animal-related experiments were approved by the Institutional Animal Care and Use Committee of MD Anderson Cancer Center. The right uterine horns of 8-week-old female athymic nude mice were injected with five million Hec1A cells in 100 µL of Hank’s Balanced Salt Solution (HyClone, Logan, UT, USA) to the uterine horn. For the second model, one million Ishikawa-Luc cells were injected into the peritoneal cavity of 6–8-week-old female mice. After eight days, mice were randomized to four groups (10 mice per group): siControl-DOPC nanoparticles (NPs), siEphA2-DOPC NPs, siControl-DOPC NPs with MK1775, and siEphA2-DOPC NPs with MK1775. The siRNA-DOPC NPs were given to mice twice a week intraperitoneally, and MK1775 (30 mg/kg) was administered daily by oral gavage. Once mice from any group became moribund, all mice were euthanized; mouse weight, tumor weight, ascites volume, and number of nodules were recorded.

### 4.10. RNA-Seq Analysis

Hec1A cells were plated in six-well plates at a density of 100,000 cells per well, in triplicate, and incubated overnight. After 24 h, cells were treated with DMSO, 1 µM ALW, 0.5 µM MK1775, or the combination of 1 µM ALW and 0.5 µM MK1775 for 8 h, after which the RNA was extracted using Direct-zol RNA Miniprep Plus kit (ZYMO Research, Irvine, CA, USA). RNA quality was determined by RNA integrity number using a Bioanalyzer (Agilent Technologies, Santa Clara, CA, USA), and the samples were shipped to Novogene (Sacramento, CA, USA) for RNA-Seq analysis on the Illumina NovaSeq 6000 platform. Downstream analysis was performed using a combination of programs, including hisat2, DEseq2, and ClusterProfiler software. Pathway analysis was performed using Ingenuity Pathway Analysis (IPA) (Qiagen, Hilden, Germany).

### 4.11. Statistical Analysis

Statistics were performed using unpaired t-tests for comparisons between two groups and one-way ANOVAs with the Tukey post hoc test for multiple comparisons between more than two groups (Prism). Statistical significance was defined as a *p* value of <0.05.

## 5. Conclusions

In conclusion, EphA2- and Wee1-targeted therapies show synergistic interaction in endometrial cancer in both in vitro and in vivo experiments. The combination of cell cycle checkpoint inhibitors and EphA2-targeted therapy may have utility in the treatment of endometrial cancer and warrants further investigation.

## Figures and Tables

**Figure 1 ijms-24-03915-f001:**
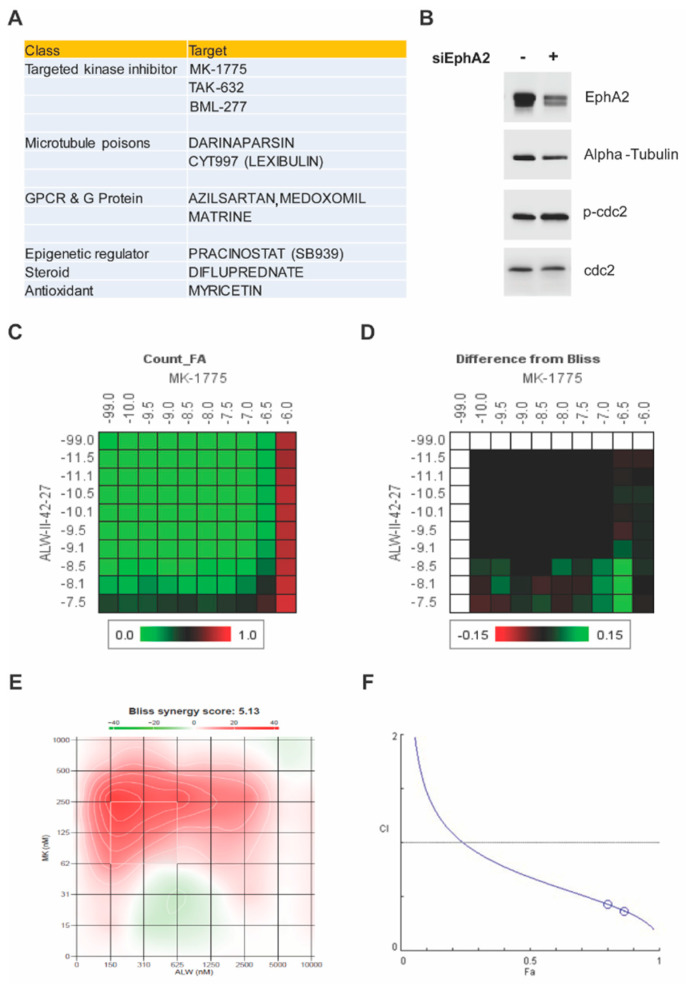
High–throughput screen for synergistic partners for EphA2–targeted therapy and verification of Wee1 as a target. (**A**) Top 10 hits from the chemical screen. (**B**) Effect of EphA2 silencing on Wee1 activity, as seen by phosphorylation of cdc2, in Hec1A cells. (**C**) Dose–response matrix for combinatorial analysis of the EphA2 inhibitor ALW and Wee1 inhibitor MK1775. (**D**) Difference from predicted Bliss independence surface model for the combination of ALW and MK1775. (**E**) Visualization of the calculated 2D synergy maps for the MTT cell viability assay from the SynergyFinder Bliss independence model combinatorial analysis. Red regions represent synergy, and green regions represent antagonism. (**F**) Plots showing fraction of cells affected (Fa) and combination index (CI) values for ALW and MK1775. The white circles representing the CI values of the ALW:MK1775 combination (1 µM:0.5 µM and 1 µM:0.75 µM concentrations, respectively) were less than one, indicating synergy.

**Figure 2 ijms-24-03915-f002:**
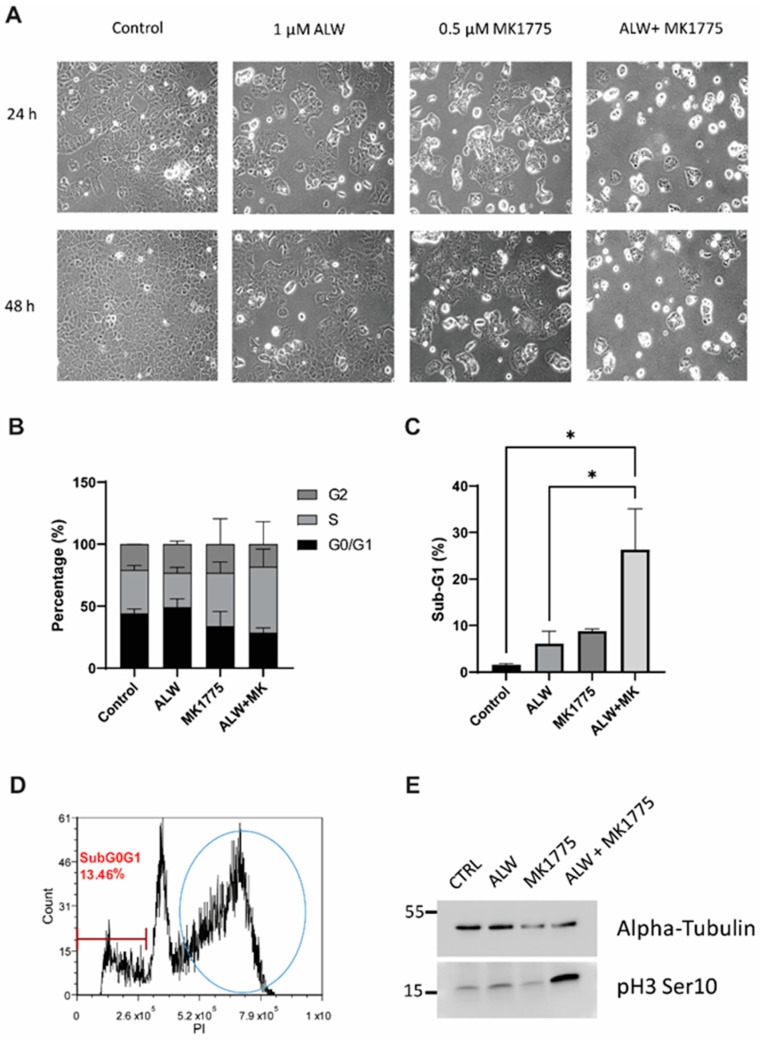
Effects of combination therapy on cell proliferation. (**A**) Cell morphology images of untreated (control) and ALW- and MK1775-treated Hec1A cells at 24 h and 48 h timepoints. (**B**,**C**) Flow cytometry analysis of cell cycle phases in Hec1A cells that were untreated or treated for 48 h. (**D**) Cell cycle profile of ALW- and MK1775-treated cells showing cells arrested between the S and G2 phases. (**E**) Western blot analysis for phosphorylated histone H3, an indicator of cells arrested in mitosis. Statistics were performed using one-way ANOVA with the Tukey’s post hoc test for multiple comparisons for more than two groups. * *p* < 0.05.

**Figure 3 ijms-24-03915-f003:**
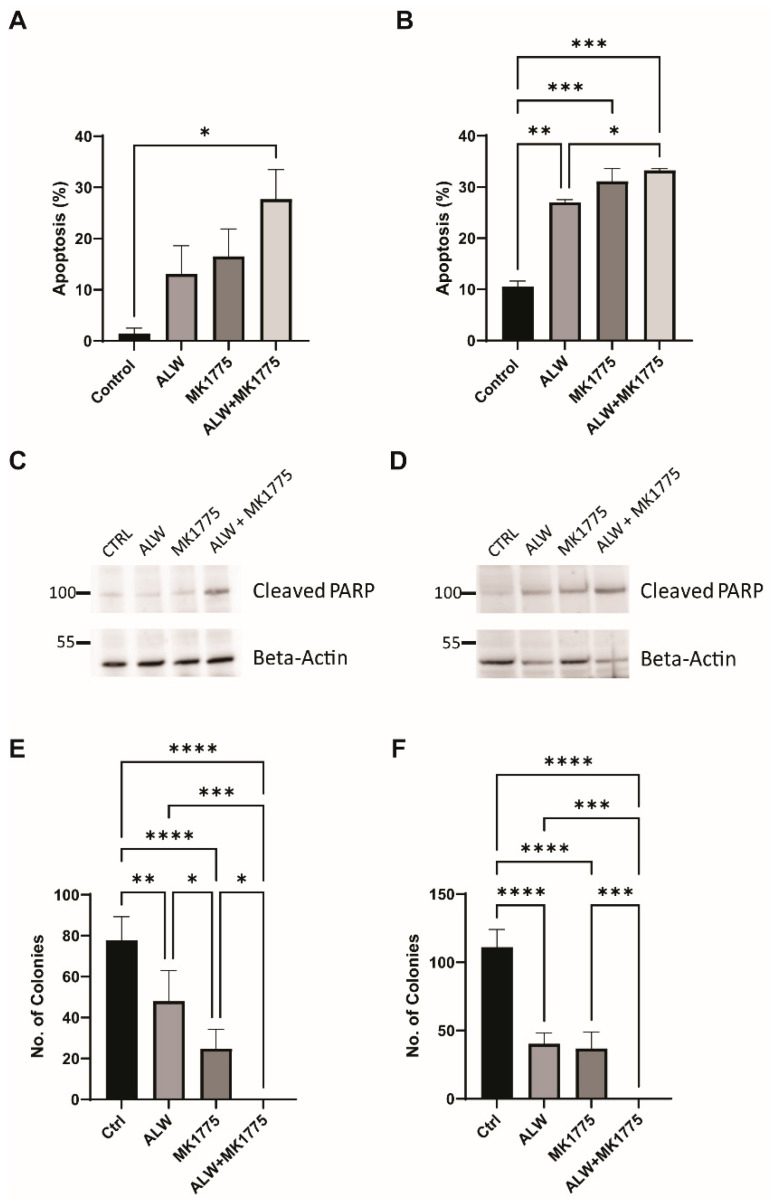
Apoptotic and clonogenic effects of combination therapy. (**A**,**B**) Flow cytometry analysis of apoptosis in Hec1A (**A**) and Ishikawa (**B**) cells that were untreated or treated with ALW, MK1775, or both drugs. (**C**,**D**) Expression of apoptosis marker cleaved PARP in untreated or drug-treated Hec1A (**C**) and Ishikawa (**D**) cells. (**E**,**F**) Clonogenic colony formation assay in untreated or drug-treated Hec1A (**E**) and Ishikawa (**F**) cells. Statistics were performed using one-way ANOVA with the Tukey’s post hoc test for multiple comparisons for more than two groups. ns, nonsignificant; * *p* < 0.05; ** *p* < 0.01; *** *p* < 0.001; **** *p* < 0.0001.

**Figure 4 ijms-24-03915-f004:**
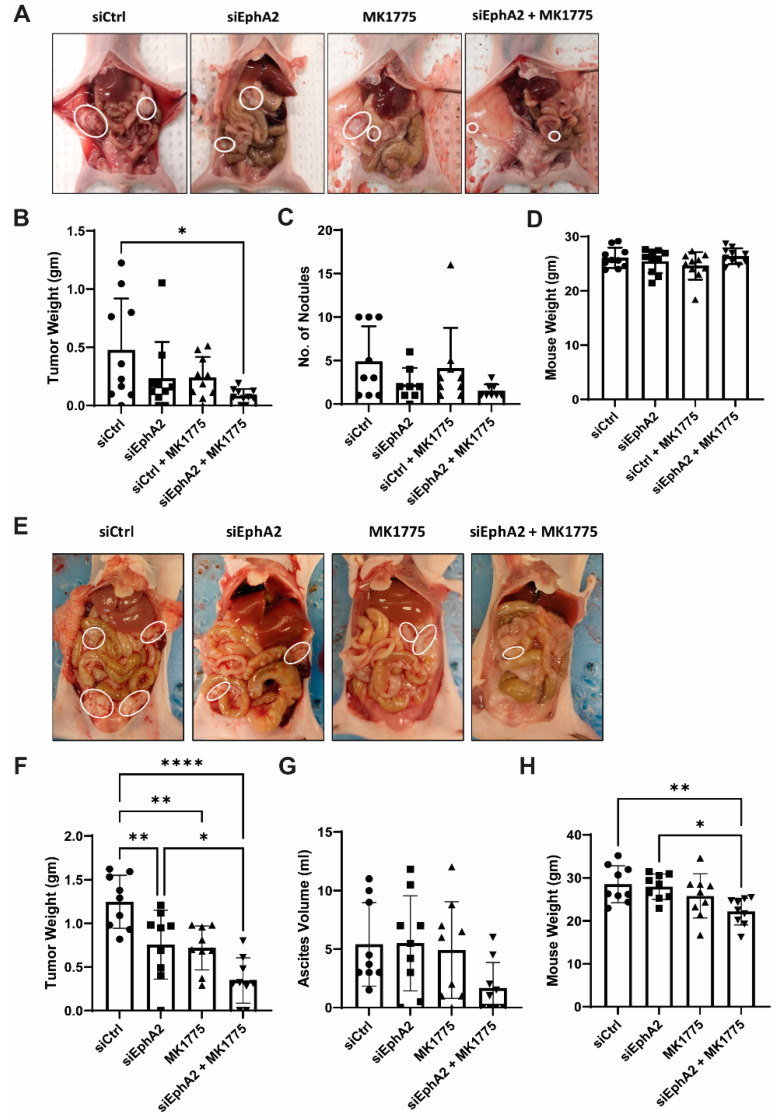
Anti-tumor effects of EphA2- and Wee1-targeted therapy in endometrial cancer xenograft models. Representative images of tumor burden in mice with Hec1A (**A**) and Ishikawa-Luc (**E**) tumors with siRNA nanoparticles and/or MK1775 therapy. (**B**,**C**) Tumor weights (**B**) and number of nodules (**C**) after therapy in Hec1A model. (**F**,**G**) Tumor weights (**F**) and ascites (**G**) in Ishikawa-Luc model. Mouse body weights at the end of the experiment in Hec1A (**D**) and Ishikawa-Luc (**H**) models. Statistics were performed using one-way ANOVA with the Tukey’s post hoc test for multiple comparisons for more than two groups. * *p* < 0.05; ** *p* < 0.01; **** *p* < 0.0001.

**Figure 5 ijms-24-03915-f005:**
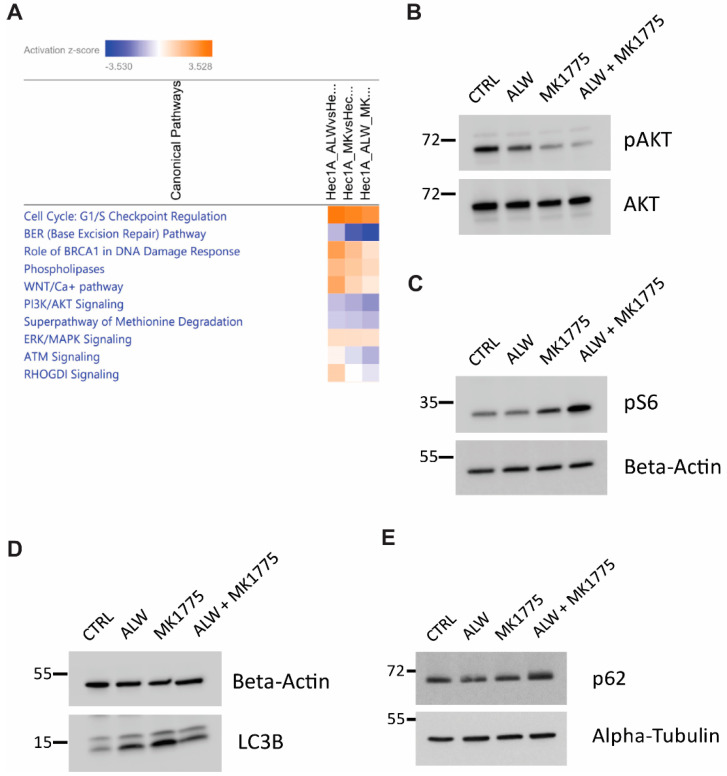
IPA analysis for identification of molecular mechanisms associated with synergy of EphA2- and Wee1–targeted therapy. (**A**) Comparative analysis of pathways downregulated in combination therapy. (**B**,**C**) Validation of RNA–Seq analysis of AKT pathway repression (**B**) and mTOR pathway activation (**C**). (**D**,**E**) Validation of RNA–Seq analysis with autophagy marker LC3B (**D**) and autophagy substrate p62 (**E**).

## Data Availability

The datasets generated during the current study are available from the corresponding author on reasonable request.

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
