# Peer review of "Combination of EphA2- and Wee1-Targeted Therapies in Endometrial Cancer"

_ijms, 2023, doi:10.3390/ijms24043915_

Round 1
Reviewer 1 Report
This paper examines candidates that enhance the effect of EphA2 using a high-throughput chemical screen to find therapeutic targets for endometrial cancer. It is very important to narrow down the drugs that can be used for treatment. Since single drugs are not effective, the use of multiple drugs may enhance the effect. It will become important in the future as one of the methods of drug discovery.
1, You found synergistic Wee1 by high-throughput chemical screen from EphA2. Would the same result be obtained if the opposite pattern was used?
Reviewer 2 Report
The authors investigated the clinical benefit of combination therapy "EphA2-targeted therapy and Wee1 inhibition therapy" using both of in vitro and in vivo experiments. Overall, it is very exciting study that shows the potential of EphA2-targeted therapy in treatment of endometrial cancer.
Question 1)
・Both of cell lines which you used are endometrioid endometrial carcinoma. How do you think about the clinical response in other histological type, such as serous carcinoma?
Question 2)
・In your citation paper, EphA2 overexpression is related to lack of hormone receptor expression and poor clinical outcome. Is the drug response affected by the presence or absence of ER or PgR?
